# Safety and Feasibility of Laparoscopic Parenchymal-Sparing Hepatectomy for Lesions with Proximity to Major Vessels in Posterosuperior Liver Segments 7 and 8

**DOI:** 10.3390/cancers15072078

**Published:** 2023-03-30

**Authors:** Hirokatsu Katagiri, Hiroyuki Nitta, Syoji Kanno, Akira Umemura, Daiki Takeda, Taro Ando, Satoshi Amano, Akira Sasaki

**Affiliations:** Department of Surgery, Iwate Medical University School of Medicine, 2-1-1 Idai-dori, Yahaba 028-3609, Iwate, Japan; hnitta@iwate-med.ac.jp (H.N.); kannos@iwate-med.ac.jp (S.K.); aumemura@iwate-med.ac.jp (A.U.); dtakeda@iwate-med.ac.jp (D.T.); antaro@iwate-med.ac.jp (T.A.); satoshia@iwate-med.ac.jp (S.A.); sakira@iwate-med.ac.jp (A.S.)

**Keywords:** laparoscopic, hepatectomy, liver resection, posterosuperior, parenchymal sparing

## Abstract

**Simple Summary:**

Improvements in perioperative management and surgical techniques have enabled laparoscopic liver resection for posterosuperior liver segments. Recent studies have reported the safety and feasibility of selected posterosuperior lesions; however, laparoscopic parenchymal-sparing hepatectomy for lesions with proximity to major vessels in posterosuperior segments has not yet been examined. The aim of this study is to examine the safety and feasibility of laparoscopic parenchymal-sparing hepatectomy for lesions with proximity to major vessels in posterosuperior segments 7 and 8. The present study demonstrated that laparoscopic parenchymal-sparing hepatectomy for lesions with proximity to major vessels in posterosuperior segments 7 and 8 is safe and feasible in a specialized center with a team experienced in laparoscopic liver surgery, and the HALS technique still plays an important role as minimally invasive liver resection. These findings suggest the possibility of taking steps to perform more advanced minimally invasive liver resections.

**Abstract:**

Laparoscopic parenchymal-sparing hepatectomy (PSH) for lesions with proximity to major vessels (PMV) in posterosuperior segments (PSS) has not yet been sufficiently examined. The aim of this study is to examine the safety and feasibility of laparoscopic PSH for lesions with PMV in PSS 7 and 8. We retrospectively reviewed the outcomes of laparoscopic liver resection (LLR) and open liver resection (OLR) for PSS lesions and focused on patients who underwent laparoscopic PSH for lesions with PMV in PSS. Blood loss was lower in the LLR group (*n* = 110) than the OLR group (*n* = 16) (*p* = 0.009), and no other short-term outcomes were significantly different. Compared to the pure LLR group (*n* = 93), there were no positive surgical margins or complications in hand-assisted laparoscopic surgery (HALS) (*n* = 17), despite more tumors with PMV (*p* = 0.009). Regarding pure LLR for one tumor lesion, any short-term outcomes in addition to the operative time were not significantly different between the PMV (*n* = 23) and no-PMV (*n* = 48) groups. The present findings indicate that laparoscopic PSH for lesions with PMV in PSS is safe and feasible in a matured team, and the HALS technique still plays an important role.

## 1. Introduction

Since laparoscopic liver resection (LLR) was first reported in the early 1990s, it has gradually spread as a minimally invasive surgery [1]. Previous studies have demonstrated that LLR results in improved short-term outcomes and comparable oncological outcomes compared with open liver resection (OLR). Currently, LLR is one of the standard treatments for anterolateral segments and left lateral sectionectomy [2,3,4,5]. However, LLR for posterosuperior liver segments (PSS; segments 1, 4b, 7, and 8) remains the most challenging procedure. According to the European Consensus Conference held in Southampton in 2017, a technically demanding resection for lesions located in PSS has yet to be fully standardized and should only be performed in specialized centers [6].

In recent years, appropriate anesthetic respiratory and circulatory management and the development of surgical techniques have enabled LLR to be performed safely at many hospitals, and recent studies have indicated that LLR is technically feasible and safe for selected patients with lesions in PSS [7,8,9,10,11,12,13,14]. Nevertheless, laparoscopic parenchymal-sparing hepatectomy (PSH) for liver lesions with proximity to major vessels (PMV) in PSS has not yet been examined. Due to the variation in the degree of difficulty of LLR, depending on the procedure of the hepatectomy and tumor conditions, a difficulty scoring system that assigns increasing values to tumors in close proximity to major vessels was proposed [15]. In PSS, especially segments 7 and 8, this factor is likely to have a greater impact on surgical outcomes.

The aim of the present study is to examine the safety and feasibility of laparoscopic PSH for lesions with PMV in PSS, especially segments 7 and 8, and to explore the possibility of taking further steps to perform minimally invasive liver resections.

## 2. Materials and Methods

### 2.1. Selection of Patients and Data Collection

A prospective database of the patients treated at our institution was retrospectively reviewed. Between January 2011 and December 2021, 1041 patients underwent liver resections at our institution. Consistently, 80–90% of cases had been treated with laparoscopic surgery during the inclusion period. During this study period, 165 patients underwent PSH for liver tumors (hepatocellular cell carcinoma, metastatic liver cancer, cholangiocellular carcinoma, and benign tumors) located in PSS 7 or 8. From this subset, we excluded 39 patients who underwent hepatectomies for four or more lesions, combined resection of other organs, hybrid technique, or resections concomitant with laparoscopic major hepatectomy. The exclusion criteria for LLR were 4 or more lesions resected, lesions spreading to other organs needed reconstruction, patients needing regional lymph node dissection, and the need for bile ducts and/or vessels resection with reconstruction. These are indicated for open surgery. Neither the size of the lesions nor cirrhosis were exclusion criteria.

A total of 126 patients (110 patients in the LLR group and 16 patients in the OLR group) who underwent PSH for lesions located in PSS 7 or 8 were retrospectively reviewed. To assess safety and feasibility within the LLR group, we divided the 110 patients in the LLR group into two groups: a group of 35 patients with lesions with PMV and a group of 75 patients with lesions with no PMV (no-PMV). To clarify the role of the hand-assisted laparoscopic surgery (HALS) technique, we reviewed 93 patients in the pure LLR group and 17 patients in the HALS group. Finally, to account for some discrepancies in the background factors, we further analyzed patients who underwent pure LLR for one tumor lesion, including 71 patients (23 patients in the PMV group and 48 patients in the no-PMV group).

The following variables were examined in our analysis: patient characteristics (age, sex, and body mass index (BMI); histories of preoperative chemotherapy, upper abdominal surgery, and hepatectomy; Child–Pugh score, and physical status score by the American Society of Anesthesiologists physical status classification (ASA-PS)); preoperative laboratory data (plasma aspartate aminotransferase (AST), plasma alanine aminotransferase (ALT), plasma total bilirubin, plasma albumin level, prothrombin time international normalized ratio (PT-INR), blood platelet count, and indocyanine green retention rate at 15 min); pathological factors (presence of liver cirrhosis, tumor number, tumor size, and location); intraoperative factors (surgical procedures, Pringle’s maneuver, operation time, volume of blood loss, blood transfusion rate, sacrifice of major hepatic veins, and positive surgical margin); and postoperative information (length of hospital stay, morbidity, and mortality). 

The present study protocol was approved by the Institutional Review Board of Iwate Medical University. All patients were informed about this study, and consent was obtained.

### 2.2. Definitions

Laparoscopic liver resection was defined as a pure laparoscopic surgery or a HALS technique. Proximity to major vessels was defined as the main or second branches of Glisson’s tree, major hepatic vein, and inferior vena cava [15] (Figure 1a,b). Postoperative morbidity was graded according to the Clavien–Dindo classification [16]. Postoperative mortality was defined as any death occurring within 90 days of liver resection. The surgical margin was defined as microscopically positive if tumor cells were identified along the periphery of the resected specimen.

### 2.3. Surgical Procedure

For segment 7 resections, the patients were treated in the left half-lateral decubitus position. For segment 8 resections, the patients were treated in the supine position. An anti-Trendelenburg position was used in all cases. The operator stood to the right of the patient, while the assistant and scopist were on the patient’s left. The anesthesiologist maintained a low central vein pressure of ≤3 mmHg and a low airway pressure ≤ 15 cm H_2_O to reduce bleeding from the hepatic vein [17]. A carbon dioxide pneumoperitoneum was maintained at 10 mmHg. Visual exploration of the abdominal cavity was conducted with a flexible endoscope. Intraoperative ultrasonography was routinely used to identify the location of the tumors and surgical boundaries and to confirm hepatic blood flow. In the HALS technique, a hand-assisted device (Wound Retractor^TM^, Applied Medical, Rancho Santa Margarita, CA, USA) was placed in the right abdominal horizontal incision (7–9 cm). The intermittent Pringle’s maneuver was continuously repeated during parenchymal transection. Trocar placement is shown in Figure 2a,b.

We mobilized the right liver from the lateral and posterior abdominal walls and created a space on the right side of the inferior vena cava for dorsal retraction during parenchymal transection. This procedure is a crucial preparation for bleeding control. Following these preparatory steps, the liver parenchyma of segments 7 and 8 were transected using the clamp crush method and a sealing device. Bleeding from small branches of the hepatic veins was controlled by a saline dripping monopolar soft-coagulation system. Instances of bleeding caused by tearing in the small crotch of vessels branches, which we call a hangnail injury, were treated by making a clean cut while the initial tear was small (Figure 3a–c). After making a clean cut, a saline dripping monopolar soft-coagulation system was used to achieve hemostasis (Figure 3d). In case of bleeding from the branches of major hepatic veins, compression of the liver parenchyma from the dorsal side, which we call dorsal compression, enabled a safe operation with controlled bleeding (Figure 4). If necessary, the hepatic vein was divided using a stapler with a 60 mm cartridge. After resection of the targeted liver tissue was performed, the specimen was extracted through an incision using a protective bag. 

### 2.4. Statistical Analysis

The continuous variables are described as medians with interquartile ranges, whereas the categorical variables are described as totals and frequencies. Differences in groups were assessed through Student’s *t*-test or ANOVA for the continuous parametric variables, Mann–Whitney U test for the continuous non-parametric variables, and Pearson’s chi-squared test or Fisher’s exact test (for expected counts of <5) for the categorical variables. Survival was estimated using the Kaplan–Meier method and compared between the groups using the log-rank test. Statistical analysis was performed using JMP software (version 13.2.0, SAS Institute, Cary, NC, USA). Variables with a *p*-value < 0.05 were considered statistically significant.

## 3. Results

### 3.1. Analysis of 126 Patients

We analyzed the data of 126 patients, of which 110 underwent LLR and 16 underwent OLR. Male patients were more common in the OLR group (LLR vs. OLR: 68.2% vs. 93.7%, *p* = 0.034). Histories of upper abdominal surgery and hepatectomy were more common in the OLR group (LLR vs. OLR: 24.5% vs. 68.7%, *p* < 0.001 and 12.7% vs. 43.7%, *p* = 0.002, respectively). Patients with liver cirrhosis were more frequent in the LLR group (LLR vs. OLR: 13.6% vs. 0.0%, *p* = 0.010). Except for the indocyanine green retention rates at 15 min (LLR vs. OLR: 13% vs. 15%, *p* = 0.030), there was no significant difference in the laboratory data. The patient characteristics are shown in Table 1.

The perioperative outcomes are shown in Table 2. Patients who underwent PSH for lesions with PMV in PSS 7 and 8 were 31.8% in the LLR group and 31.3% in the OLR group (*p* = 0.963). The major hepatic vein of four patients in the LLR group was sacrificed to remove malignant tumors (LLR vs. OLR: 3.6% vs. 0.0%, *p* = 0.438). No patients were converted to open surgery in the LLR group. Median blood loss was significantly lower in the LLR group (LLR vs. OLR: 54 mL vs. 226 mL, *p* = 0.009). Pringle’s maneuver was performed less frequently in the OLR group (LLR vs. OLR: 84.6% vs. 37.5%, *p* < 0.001). The median maximum tumor diameter and surgical margin were larger in the LLR group (LLR vs. OLR: 25.5 mm vs. 22.5 mm, *p* = 0.016, and 3.5 mm vs. 2.0 mm, *p* = 0.010, respectively). The positive surgical margin rate was higher in the OLR group (LLR vs. OLR: 4.5% vs. 18.7%, *p* = 0.030). There was no significant difference between the two groups in the median operative time (LLR vs. OLR: 205 min vs. 195 min, *p* = 0.557), median number of tumors (LLR vs. OLR: 1 vs. 1, *p* = 0.205), and morbidity rate (LLR vs. OLR: 8.1% vs. 18.7%, *p* = 0.178). Clavien–Dindo ≥ 3 morbidities were due to two surgical site infections (SSIs) and one bile leakage in the OLR group, and three SSIs, three bile leakages, two pleural fluids, and one portal vein thrombosis in the LLR group. There was no mortality within 90 days after the hepatectomy.

The median follow-up period was 780 (7–4018) days in the LLR group and 1310 (50–2869) days in the OLR group. Although the primary source of malignancy may have differed, there was no significant difference in the 5-year overall survival rate between the two groups (LLR vs. OLR: 73.9% vs. 74.3%, *p* = 0.943; Figure 5). 

### 3.2. Analysis of 110 Patients Who Underwent Laparoscopic PSH for Lesions in PSS 7 and 8 with and with No PMV

To assess safety and feasibility within the LLR group, the 110 LLR patients were divided into two groups—35 patients in the PMV group and 75 patients in the no-PMV group—and reviewed. The patient characteristics are shown in Table 3. Both of the patients with Child–Pugh B score were included in the PMV group. Statistically, plasma levels of albumin, AST, and PT-INR were significantly different between the two groups. 

The perioperative outcomes are shown in Table 4. In the PMV group, the HALS technique was performed more often (PMV vs. no-PMV: 28.6% vs. 9.3%, *p* = 0.009). The median operative time was significantly longer in the PMV group (PMV vs. no-PMV: 237 min vs. 185 min, *p* = 0.030). All four patients whose major hepatic veins were sacrificed were included in the PMV group. The median tumor diameter was larger in the PMV group (PMV vs. no-PMV: 36.0 mm vs. 24.0 mm, *p* < 0.001). The median surgical margin of the specimens was smaller in the PMV group (PMV vs. no-PMV: 3.0 mm vs. 5.0 mm, *p* = 0.010); however, there was no significant difference in the rate of positive surgical margin between the two groups. 

### 3.3. Analysis of Patients Who Underwent Pure Laparoscopic and HALS PSH in PSS 7 and 8 for Lesions with and with No PMV 

To clarify the role of the HALS technique, we reviewed 93 patients in the pure LLR group and 17 patients in the HALS group. The patient characteristics are shown in Table 5. Histories of upper abdominal surgery and hepatectomy were significantly higher in the HALS group (pure LLR vs. HALS: 21.5% vs. 41.2%, *p* = 0.039, and 9.7% vs. 29.4%, *p* = 0.025, respectively). Statistically, plasma levels of albumin and the indocyanine green retention rates at 15 min were significantly different between the two groups.

As shown in Table 6, there were no significant differences in short-term outcomes, although lesions with PMV were significantly more resected using the HALS technique (*p* = 0.009). Moreover, there were no positive surgical margins or complications in the HALS group, and there were no significant differences (pure LLR vs. HALS: 5.4% vs. 0.0%, *p* = 0.328, and 9.7% vs. 0.0%, *p* = 0.181, respectively).

### 3.4. Analysis of 71 Patients Who Underwent Pure Laparoscopic PSH for One Tumor Lesion with PMV in PSS 7 and 8

To account for some background factors, we performed a further analysis of patients who underwent pure laparoscopic PSH (pPSH) for one tumor lesion with PMV in PSS 7 and 8, including 71 patients (23 patients in the pPSH-PMV group and 48 patients in the pPSH-no-PMV group). The patient characteristics are shown in Table 7. Child–Pugh B patients were included only in the pPSH-PMV group. Statistically, the plasma levels of albumin, AST, ALT, and PT-INR were significantly different between the two groups. There were no significant differences in any other variables between the two groups.

As shown in Table 8, the perioperative outcomes were compared between the two groups. All of the three patients whose major hepatic veins were sacrificed were included in the pPSH-PMV group. The median tumor diameter was significantly larger in the pPSH-PMV group (pPSH-PMV vs. pPSH-no-PMV: 36.0 mm vs. 23.0 mm, *p* < 0.001). The median operative time was significantly different between the two groups (pPSH-PMV vs. pPSH-no-PMV: 240 min vs. 163 min, *p* = 0.002). The median surgical margin of the specimens was smaller in the pPSH-PMV group (pPSH-PMV vs. pPSH-no-PMV: 3.0 mm vs. 5.0 mm, *p* = 0.008); however, there was no significant difference in the rate of positive surgical margin between the two groups (pPSH-PMV vs. pPSH-no-PMV: 8.3% mm vs. 3.8%, *p* = 0.310). The median blood loss, length of hospital stay, and Clavien–Dindo ≥ 3 morbidities were not significantly different between the two groups (pPSH-PMV vs. pPSH-no-PMV: 98 mL vs. 50 mL, *p* = 0.364; 10 days vs. 9 days, *p* = 0.654, and 11.1% vs. 6.4%, *p* = 0.387, respectively).

## 4. Discussion

The lesions located in PSS 7 and 8 were considered difficult to address using LLR due to limited visualization, restrictions on surgical manipulation, and their proximity to major hepatic veins. At the Consensus Conference held at Morioka in 2014, the jury concluded that PSH for PSS was not a minor operation and was still considered an innovative procedure [5]. According to expert recommendations, LLR for lesions with PMV is not contraindicated to be performed in a specialized center [18]. Recent international consensus meetings held in Southampton recommended that PSH for PSS be performed by experienced surgeons in select patients in high-volume centers [6]. Some comparative studies have reported that, for tumors in PSS, LLR is superior to OLR in terms of intraoperative blood loss, postoperative hospital stay, and major complication rates [7,8,9,10,11,12,13]. In the present study, we showed that LLR for lesions in PSS resulted in lower intraoperative blood loss than OLR and that there was no difference in short-term and long-term outcomes between the two groups. However, these results should be carefully interpreted because some selection biases possibly exist in both groups despite our criteria for open or laparoscopic surgery. Moreover, we cannot assert that there are no differences in the long-term results because various types of tumors were included in this study. With regards to the cholangiocellular carcinoma in this study in PSS 7 and 8, a regional hepatic hilum lymphadenectomy was not performed because the main legion was located far from the hepatic hilum.

Some tumors in PSS 7 and 8 can be closer in proximity to major hepatic veins than others. Laparoscopic major hepatectomies have frequently been performed for lesions with PMV in PSS [19]. A previous report demonstrated that PSH for colorectal liver metastases (CRLM) has been associated with decreased mortality and equivalent survival compared to major hepatectomies [20]. Another report showed that laparoscopic minor hepatectomies for lesions in PSS showed no statistical difference in blood loss or operation time [21]. We examined the safety and feasibility of laparoscopic PSH for lesions with PMV in PSS because the proximity to the major hepatic veins is likely to have a greater impact on short-term surgical outcomes. This study shows that laparoscopic PSH for lesions with PMV in PSS 7 and 8 remains safe and feasible in terms of short-term outcomes. The operative time in the pPSH-PMV group was longer than in the pPSH-no-PMV group. However, there was no difference in intraoperative blood loss, complication rates, or postoperative hospital stay between the two groups, despite the larger tumor diameters in the pPSH-PMV group compared to the pPSH-no-PMV group. Although an appropriate surgical margin might not be adequately guaranteed when performing PSH for lesions with PMV in PSS, our results demonstrated that there was no difference in the positive surgical margin rate between the two groups. Frequent intraoperative ultrasound may contribute to securing the margin [13].

A previous study that compared the HALS technique to OLR for CRLM in PSS demonstrated that the HALS technique is a safe, feasible, and preferable approach because it leads to a lower complication rate and shorter hospital stays without compromising survival and disease recurrence [7]. We adopted the HALS technique for patients with tumors located in close proximity to major hepatic veins in PSS that would likely have required sacrificing the major hepatic veins in preoperative estimations. Although various types of tumors were included, there were tendencies toward lower positive surgical margins and lower complication rates in the HALS group. The HALS technique for lesions with PMV in PSS is useful because it permits a good view of the operation field during parenchymal transection and controls bleeding when performing continuous dorsal compression. In addition, it is oncologically useful to detect multiple small superficial lesions and ensure safe surgical margins, especially for borderline invasive tumors, such as CRLMs, using tactile sensation. Currently, we consider that the HALS technique still plays an important role as a minimally invasive form of liver resection and not simply a bridge between open and laparoscopic surgeries.

Good visibility, adequate preparation for bleeding control, and effective management in the event of bleeding are important factors in safely performing LLR for tumors in PSS [22]. Some successive approaches to overcoming poor surgical manipulation or operating views in PSS have been reported [23,24,25,26,27,28]. We believe that full mobilization of the right side of the liver is essential for obtaining a better view of the operating field, controlling bleeding, and performing LLR for lesions with PMV in PSS. Fortunately, we were able to obtain a good visualization of the operation field and perform LLR without stress by using a flexible scope with almost the same port placement for lesions in any segment (Figure 2a,b). It should be emphasized that the experience and skill of a scopist is as important as the operator’s proficiency. Moreover, full mobilization of the right side of the liver becomes a crucial preparation for bleeding control. We propose that creating a space on the right side of the inferior vena cava and compressing the liver parenchyma from the dorsal side (dorsal compression) enable a safe operation with controlled bleeding [29] (Figure 4). In addition to pneumoperitoneum and anesthesia management, it is highly beneficial that the force of gravity can be used for hemostasis by positioning the main blood vessels on the dorsal side of the resection plane (left half-lateral decubitus position l for segment 7 and supine position for segment 8) [30]. Dorsal compression also enables subtle adjustment to maintain a bleeding point ventral to such major vessels as the right hepatic vein and the inferior vena cava. The most important hemostatic management is probably the hangnail injury described above. Attempting to treat blindly or applying too much tension for visualization could result in tearing of the major vessels, leading to hemorrhage and irreparable damage. It is crucial to cleanly cut away a hangnail injury, rather than tearing it away while the initial tear is small. After making a clean cut, a saline dripping monopolar soft-coagulation system facilitates hemostasis using many of the aforementioned management methods (Figure 3a–d). The application of our useful surgical procedures and techniques may enable the achievement of a good view of the operation field, preparation for controlling bleeding, and management in the event of bleeding, while providing significant benefits for safety and feasibility.

Although pneumoperitoneal pressure is highly beneficial for bleeding control when performing laparoscopic PSH with PMV in PSS, caution regarding paradoxical carbon dioxide embolism should always be exercised [31]. Operating close to the major vessels carries the risk of damaging the major hepatic vein. Even if the bleeding is controlled successfully, it is important to keep in mind the possibility of cerebral infarction by paradoxical gas embolism and to be prepared for treatment by methods such as discontinuing pneumoperitoneum, changing the patient’s position from head-up to head-down, or closure of the injured vein either directly or by dorsal compression.

This study has certain limitations. First, this study was not a prospective or a randomized study but a retrospective design. Second, we could not perform effective statistical analyses due to the small sample size. Third, we were unable to sufficiently examine the long-term results of pure LLR for each type of cancer disease due to the short follow-up period. Fourth, this study did not show any differences in hospital stays, which may be explained by Japan’s national health insurance. Although the lengths of hospital stays are mainly determined by physicians’ clinical judgment, patients and their family members often participate in determining discharge dates. It may be difficult to compare the lengths of hospital stays in Japan with those in other countries. Finally, a few patients underwent open PSH for liver tumors located in PSS 7 or 8 because most of the single-tumor lesions in these segments were resected laparoscopically in our institution. 

## 5. Conclusions

The present findings indicate that laparoscopic PSH for lesions with PMV in PSS, especially segments 7 and 8, is safe and feasible in high-volume specialized centers with a team experienced in laparoscopic liver surgery. Nevertheless, it will be necessary to consider and estimate the surgeon’s experience and well-selected indications in the future. Furthermore, the HALS technique still plays an important role as a minimally invasive liver resection method, beyond being a mere bridge between open and laparoscopic surgeries. 

## Figures and Tables

**Figure 1 cancers-15-02078-f001:**
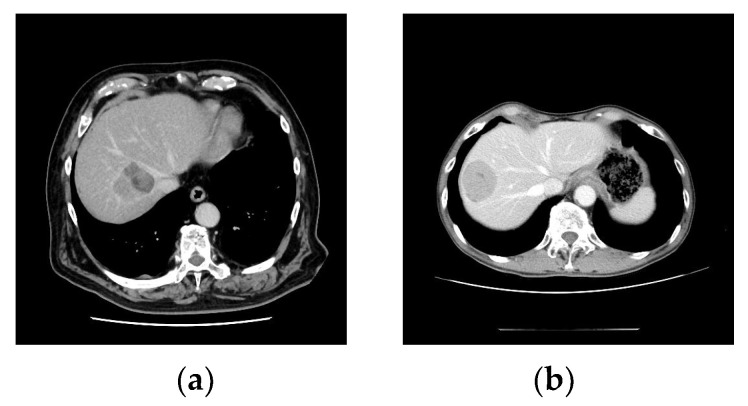
(**a**) A lesion with proximity to major vessels. (**b**) A lesion with no proximity to major vessels.

**Figure 2 cancers-15-02078-f002:**
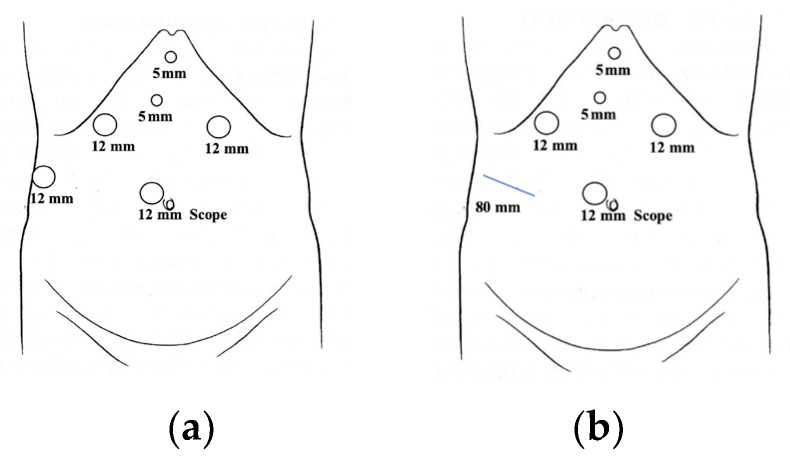
(**a**) Trocar placement for pure LLR. (**b**) Trocar placement for HALS. The blue line is the incision placed by a hand-assisted device.

**Figure 3 cancers-15-02078-f003:**
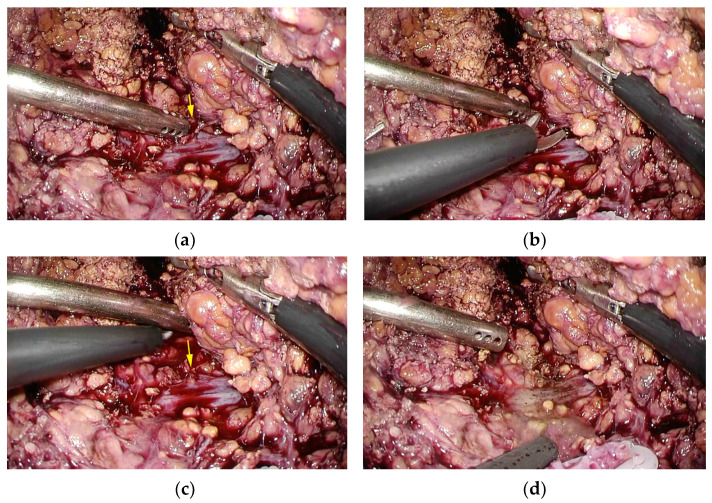
(**a**) The small crotch formed teared from vessels. The yellow arrow is the hangnail injury. (**b**) Sharply cutting the small branch teared from vessels. (**c**) Releasing the tension against the vessels. The yellow arrow is the hangnail injury. (**d**) The hemostasis using a saline dripping monopolar soft-coagulation system.

**Figure 4 cancers-15-02078-f004:**
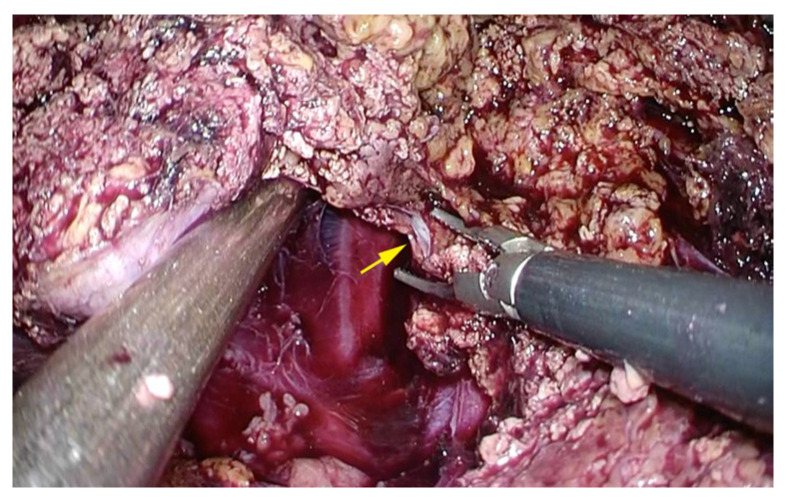
Dorsal compression, compressing the liver parenchyma from the dorsal side, enables control of bleeding from a branch of the hepatic vein. The yellow arrow is the bleeding point from the hepatic vein.

**Figure 5 cancers-15-02078-f005:**
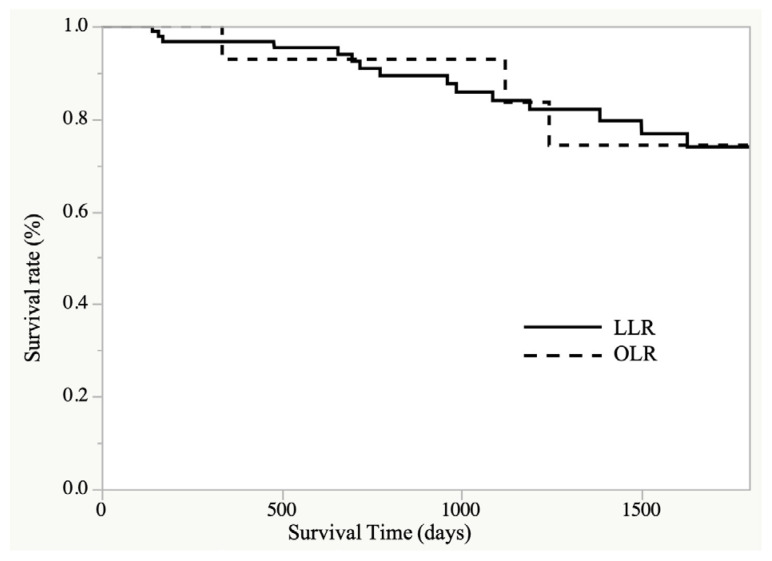
Overall survival after LLR and OLR groups.

**Table 1 cancers-15-02078-t001:** Characteristics of all patients that underwent PSH for lesions in PSS 7 and 8.

	LLR (*n* = 110)	OLR (*n* = 16)	*p*-Value
Sex (male)	75 (68.2%)	15 (93.7%)	0.034 *
Age (years)	68 (25–85)	72 (61–83)	0.065
ASA-PS			0.254
1	26 (23.6%)	1 (6.2%)	
2	66 (60.0%)	11 (68.8%)	
3	18 (16.4%)	4 (25.0%)	
BMI (kg/m^2^)	23.7 (16.1–35.2)	23.2 (17.1–29.7)	0.371
Histories of upper abdominal surgery	27 (24.5%)	11 (68.7%)	<0.001 *
Histories of hepatectomy	14 (12.7%)	7 (43.7%)	0.002 *
Preoperative chemotherapy	28 (25.5%)	3 (18.7%)	0.561
Child–Pugh B	2 (1.8%)	0 (0.0%)	0.586
Cirrhosis	15 (13.6%)	0 (0.0%)	0.010 *
Laboratory data			
Albumin (g/dL)	4.1 (3.1–5.0)	4.1 (3.5–4.6)	0.788
AST (IU/L)	24 (12–146)	26 (12–174)	0.447
ALT (IU/L)	22 (6–218)	23 (8–184)	0.303
Total bilirubin (mg/dL)	0.6 (0.2–1.9)	0.6 (0.3–1.1)	0.697
PT-INR	1.03 (0.85–1.43)	1.02 (0.94–1.24)	0.897
Platelet count (10^3^/μL)	180 (64–702)	170 (95–325)	0.885
ICG-R15 (%)	13 (2–53)	15 (3–44)	0.030 *

Data are shown as median (range) or number of cases (percentage). *: statistically significant. LLR, laparoscopic liver resection; OLR, open liver resection; ASA-PS, American Society of Anesthesiologists physical status; BMI, body mass index; AST, aspartate aminotransferase; ALT, alanine aminotransferase; PT-INR, prothrombin time international normalized ratio; ICG-R15, indocyanine green retention rates at 15 min.

**Table 2 cancers-15-02078-t002:** Perioperative outcomes of LLR and OLR groups.

	LLR (*n* = 110)	OLR (*n* = 16)	*p*-Value
Pathological Diagnosis			0.029 *
HCC	44 (40.0%)	5 (31.2%)	
CRLM	49 (44.6%)	1 (6.3%)	
CCC	0 (0.0%)	8 (50.0%)	
Other malignancy	11 (10.0%)	2 (12.5%)	
Benign	6 (5.4%)	0 (0.0%)	
Surgical procedure of LLR			NA
Pure	93 (84.5%)	NA	
HALS	17 (15.5%)	NA	
Operative time (minutes)	205 (66–710)	195 (131–338)	0.557
Blood loss (mL)	54 (1–3026)	226 (90–2880)	0.009 *
Blood transfusion	2 (1.8%)	1 (6.3%)	0.277
Pringle’s maneuver	93 (84.6%)	6 (37.5%)	<0.001 *
Lesions with proximity to major vessels	35 (31.8%)	5 (31.3%)	0.963
Sacrifice of major hepatic veins	4 (3.6%)	0 (0.0%)	0.438
Largest tumor diameter (mm)	25.5 (7.0–110.0)	22.5 (10.0–170.0)	0.016 *
Number of tumors	1 (1–3)	1 (1–3)	0.205
Surgical margin (mm)	3.5 (0.0–25.0)	2.0 (0.0–9.0)	0.010 *
Positive surgical margin	5 (4.5%)	3 (18.7%)	0.030 *
Length of hospital stay (days)	10 (4–158)	13 (7–110)	0.129
Morbidity (Clavien–Dindo ≥ 3)	9 (8.1%)	3 (18.7%)	0.178
Mortality	0 (0.0%)	0 (0.0%)	NA

Data are shown as median (range) or number of cases (percentage). *: statistically significant. LLR, laparoscopic liver resection; OLR, open liver resection; HCC, hepatocellular carcinoma; CRLM, colorectal liver metastases; CCC, cholangiocellular carcinoma; LLR, laparoscopic liver resection; HALS, hand-assisted laparoscopic surgery; NA, not applicable.

**Table 3 cancers-15-02078-t003:** Characteristics of patients that underwent laparoscopic PSH for lesions in PSS 7 and 8.

	PMV (*n* = 35)	no-PMV (*n* = 75)	*p*-Value
Sex (male)	24 (68.6%)	51 (68.0%)	0.952
Age (years)	70 (25–83)	68 (27–85)	0.879
ASA-PS			0.828
1	7 (20.0%)	19 (25.3%)	
2	22 (62.9%)	44 (58.7%)	
3	6 (17.1%)	12 (16.0%)	
BMI (kg/m^2^)	25.3 (19.2–35.1)	23.5 (16.1–35.2)	0.064
Histories of upper abdominal surgery	8 (22.8%)	19 (25.3%)	0.878
Histories of hepatectomy	4 (11.4%)	10 (13.3%)	0.78
Preoperative chemotherapy	5 (14.3%)	23 (30.7%)	0.066
Child–Pugh B	2 (5.7%)	0 (0.0%)	0.036 *
Cirrhosis	6 (17.1%)	9 (12.0%)	0.464
Laboratory data			
Albumin (g/dL)	4.0 (3.1–4.6)	4.1 (3.4–5.0)	0.006 *
AST (IU/L)	30 (13–146)	22 (12–141)	0.023 *
ALT (IU/L)	26 (8–102)	20 (6–218)	0.264
Total bilirubin (mg/dL)	0.6 (0.2–1.9)	0.5 (0.3–1.6)	0.344
PT-INR	1.04 (0.90–1.34)	1.02 (0.85–1.43)	0.018 *
Platelet count (10^3^/μL)	163 (73–339)	186 (64–702)	0.238
ICG-R15 (%)	13 (3–33)	13 (2–53)	0.143

Data are shown as median (range) or number of cases (percentage). *: statistically significant. PMV, lesions with proximity to major vessels; no-PMV, lesions with no proximity to major vessels; ASA-PS, American Society of Anesthesiologists physical status; BMI, body mass index; AST, aspartate aminotransferase; ALT, alanine aminotransferase; PT-INR, prothrombin time international normalized ratio; ICG-R15, indocyanine green retention rates at 15 min.

**Table 4 cancers-15-02078-t004:** Surgical outcomes of PMV and no-PMV groups.

	PMV (*n* = 35)	no-PMV (*n* = 75)	*p*-Value
Pathological Diagnosis			0.146
HCC	20 (57.1%)	24 (32.0%)	
CRLM	10 (28.6%)	39 (52.0%)	
Other malignancy	2 (5.7%)	9 (12.0%)	
Benign	3 (8.6%)	3 (4.0%)	
Surgical procedure of LLR			0.009 *
Pure	25 (71.4%)	68 (90.7%)	
HALS	10 (28.6%)	7 (9.3%)	
Operative time (minutes)	237 (105–397)	185 (66–710)	0.030 *
Blood loss (mL)	121 (10–1942)	50 (1–3026)	0.116
Blood transfusion	1 (2.9%)	1 (1.3%)	0.577
Pringle’s maneuver	31 (88.6%)	62 (82.7%)	0.425
Sacrifice of major hepatic veins	4 (11.4%)	0 (0.0%)	0.003 *
Largest tumor diameter (mm)	36.0 (7.0–110.0)	24.0 (8.0–80.0)	<0.001 *
Number of tumors	1 (1–3)	1 (1–3)	0.033 *
Surgical margin (mm)	3.0 (0.0–13.0)	5.0 (0.0–25.0)	0.010 *
Positive surgical margin	2 (5.7%)	3 (4.0%)	0.687
Length of hospital stay (days)	10 (5–50)	9 (4–158)	0.721
Morbidity (Clavien–Dindo ≥ 3)	4 (11.4%)	5 (6.7%)	0.396

Data are shown as median (range) or number of cases (percentage). *: statistically significant. PMV, lesions with proximity to major vessels; no-PMV, lesions with no proximity to major vessels; HCC, hepatocellular carcinoma; CRLM, colorectal liver metastases; HALS, hand-assisted laparoscopic surgery.

**Table 5 cancers-15-02078-t005:** Characteristics of patients that underwent pure laparoscopic and HALS PSH for lesions in PSS 7 and 8.

	Pure LLR (*n* = 93)	HALS (*n* = 17)	*p*-Value
Sex (male)	61 (65.6%)	14 (82.4%)	0.173
Age (years)	69 (25–85)	65 (48–83)	0.729
ASA-PS			0.987
1	22 (23.7%)	4 (23.5%)	
2	56 (60.2%)	10 (58.8%)	
3	15 (16.1%)	3 (17.7%)	
BMI (kg/m^2^)	23.5 (16.1–35.2)	24.5 (19.9–35.1)	0.099
Histories of upper abdominal surgery	20 (21.5%)	7 (41.2%)	0.039 *
Histories of hepatectomy	9 (9.7%)	5 (29.4%)	0.025 *
Preoperative chemotherapy	25 (26.9%)	3 (17.7%)	0.422
Child–Pugh B	2 (2.2%)	0 (0.0%)	0.542
Cirrhosis	13 (14.0%)	2 (11.8%)	0.807
Laboratory data			
Albumin (g/dL)	4.1 (3.1–5.0)	3.9 (3.4–4.3)	0.018 *
AST (IU/L)	25 (12–146)	23 (17–141)	0.537
ALT (IU/L)	21 (6–102)	22 (8–218)	0.367
Total bilirubin (mg/dL)	0.6 (0.2–1.6)	0.6 (0.3–1.9)	0.163
PT-INR	1.03 (0.85–1.34)	1.04 (0.90–1.43)	0.187
Platelet count (10^3^/μL)	180 (64–702)	160 (73–339)	0.648
ICG-R15 (%)	12 (2–53)	16 (6–33)	0.008 *

Data are shown as median (range) or number of cases (percentage). *: statistically significant. LLR, laparoscopic liver resection; HALS, hand-assisted laparoscopic surgery; ASA-PS, American Society of Anesthesiologists physical status; BMI, body mass index; AST, aspartate aminotransferase; ALT, alanine aminotransferase; PT-INR, prothrombin time international normalized ratio; ICG-R15, indocyanine green retention rates at 15 min.

**Table 6 cancers-15-02078-t006:** Surgical outcomes of pure LLR and HALS.

	Pure LLR (*n* = 93)	HALS (*n* = 17)	*p*-Value
Pathological Diagnosis			0.891
HCC	36 (38.7%)	8 (47.1%)	
CRLM	43 (46.2%)	6 (35.3%)	
Other malignancy	9 (9.7%)	2 (11.7%)	
Benign	5 (5.4%)	1 (5.9%)	
Lesions with proximity to major vessels	25 (26.9%)	10 (58.8%)	0.009 *
Operative time (minutes)	203 (66–710)	211 (138–339)	0.831
Blood loss (mL)	52 (1–3026)	110 (28–881)	0.667
Blood transfusion	2 (2.2%)	0 (0.0%)	0.542
Pringle’s maneuver	81 (87.1%)	12 (70.6%)	0.083
Sacrifice of major hepatic veins	3 (3.2%)	1 (5.9%)	0.591
Number of tumors	1 (1–3)	1 (1–3)	0.678
Largest tumor diameter (mm)	25.0 (7.0–110.0)	33.0 (8.0–80.0)	0.135
Surgical margin (mm)	4.0 (0.0–25.0)	3.0 (1.0–10.0)	0.237
Positive surgical margin	5 (5.4%)	0 (0.0%)	0.328
Length of hospital stay (days)	9 (4–158)	10 (6–18)	0.452
Morbidity (Clavien–Dindo ≥ 3)	9 (9.7%)	0 (0.0%)	0.181

Data are shown as median (range) or number of cases (percentage). *: statistically significant. LLR, laparoscopic liver resection; HALS, hand-assisted laparoscopic surgery; HCC, hepatocellular carcinoma; CRLM, colorectal liver metastases.

**Table 7 cancers-15-02078-t007:** Characteristics of patients that underwent pure laparoscopic PSH for one tumor lesion with PMV in PSS 7 and 8.

	pPSH-PMV (*n* = 23)	pPSH-no-PMV (*n* = 48)	*p*-Value
Sex (male)	15 (65.2%)	29 (60.4%)	0.696
Age (years)	72 (25–83)	69 (27–85)	0.567
ASA-PS			0.853
1	4 (17.4%)	10 (20.8%)	
2	14 (60.9%)	30 (62.5%)	
3	5 (21.7%)	8 (16.7%)	
BMI (kg/m^2^)	24.5 (19.2–31.4)	23.6 (16.1–35.2)	0.588
Histories of upper abdominal surgery	6 (26.1%)	11 (22.9%)	0.847
Histories of hepatectomy	2 (8.7%)	5 (10.4%)	0.82
Preoperative chemotherapy	2 (8.7%)	12 (25.0%)	0.106
Child–Pugh B	2 (8.7%)	0 (0.0%)	0.038 *
Cirrhosis	5 (21.7%)	8 (16.7%)	0.605
Laboratory data			
Albumin (g/dL)	4.0 (3.1–4.4)	4.2 (3.6–5.0)	0.001 *
AST (IU/L)	36 (13–146)	22 (13–80)	0.024 *
ALT (IU/L)	27 (9–102)	20 (9–100)	0.029 *
Total bilirubin (mg/dL)	0.6 (0.2–1.1)	0.6 (0.3–1.6)	0.593
PT-INR	1.05 (0.98–1.34)	1.02 (0.85–1.28)	0.006 *
Platelet count (10^3^/μL)	167 (73–331)	181 (64–702)	0.167
ICG-R15 (%)	11 (3–26)	12 (3–53)	0.829

Data are shown as median (range) or number of cases (percentage). *: statistically significant. pPSH, pure laparoscopic parenchymal-sparing hepatectomy; PMV, lesions with proximity to a major vessel; no-PMV, lesions with no proximity to a major vessel; ASA-PS, American Society of Anesthesiologists physical status; BMI, body mass index; AST, aspartate aminotransferase; ALT, alanine aminotransferase; PT-INR, prothrombin time international normalized ratio; ICG-R15, indocyanine green retention rates at 15 min.

**Table 8 cancers-15-02078-t008:** Surgical outcomes of pPSH-PMV and pPSH-no-PMV groups.

	pPSH-PMV (*n* = 23)	pPSH-no-PMV (*n* = 48)	*p*-Value
Pathological Diagnosis			0.283
HCC	15 (65.2%)	18 (37.5%)	
CRLM	5 (21.7%)	20 (41.7%)	
Other malignancy	1 (4.4%)	7 (14.6%)	
Benign	2 (8.7%)	3 (6.2%)	
Operative time (minutes) *	240 (105–397)	163 (66–710)	0.002 *
Blood loss (mL) *	98 (10–1942)	50 (90–3026)	0.364
Blood transfusion	1 (4.4%)	1 (2.1%)	0.589
Pringle’s maneuver	22 (95.6%)	38 (79.2%)	0.072
Sacrifice of major hepatic veins	3 (13.0%)	0 (0.0%)	0.011 *
Largest tumor diameter (mm) *	36.0 (10.0–110.0)	23.0 (8.0–55.0)	<0.001 *
Surgical margin (mm) *	3.0 (0.0–13.0)	5.0 (0.0–25.0)	0.008 *
Positive surgical margin	3 (8.3%)	3 (3.8%)	0.31
Length of hospital stay (days) *	10 (5–50)	9 (7–110)	0.654
Morbidity (Clavien–Dindo ≥ 3)	4 (11.1%)	5 (6.4%)	0.387

Data are shown as median (range) or number of cases (percentage). *: statistically significant. pPSH, pure laparoscopic parenchymal-sparing hepatectomy; PMV, lesions with proximity to a major vessel; no-PMV, lesions with no proximity to a major vessel; HCC, hepatocellular carcinoma; CRLM.

## Data Availability

The data presented in this study are available on request from the corresponding author.

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
