# Peer review of "Safety and Feasibility of Laparoscopic Parenchymal-Sparing Hepatectomy for Lesions with Proximity to Major Vessels in Posterosuperior Liver Segments 7 and 8"

_cancers, 2023, doi:10.3390/cancers15072078_

Round 1
Reviewer 1 Report
Congratulations for the authors in writing this paper in an interesting area of treatment of liver malignancy. The quality of paper is very good. However a minor review is request .
In the results : the title of paragraph 3.2
Analysis of 110 patients who underwent laparoscopic PSH for lesions with PMV in PSS 7 and 8 is not correct. It should be : Analysis of 110 patients who underwent laparoscopic PSH for lesions in PSS 7 and 8 with and with no PMV The title of paragraph 3.3 :Analysis of patients who underwent pure laparoscopic and HALS PSH for lesions with PMV in PSS 7 and 8 is not correct, it should be: Analysis of patients who underwent pure laparoscopic and HALS PSH in PSS 7 and 8 for lesions with and with no PMV
Table n.6 should be reviewed
Reviewer 2 Report
The study has a clear objective, which is well stated and can be considered relevant. In my opinion, the authors succeed in demonstrating that parenchyma-sparing laparoscopic resection of focal hepatic lesions, close to large vessels, located in segments 7 and 8, is possible, obtaining short-term results similar to those of open surgery.
Obviously, for this it is mandatory that the surgical team have the necessary experience and an specific equipment.
However, some of the statements presented in the conclusions deserve some comments:
-It is true that blood loss is lower in the laparoscopic group, but this could be explained by the more frequent use of the Pringle maneuver. On the other hand, although the mean size of the lesions is greater in the laparoscopic group, the largest lesions are found in the open surgery group (25.5 mm(7-110) vs 22.5 (10-170) , in which there is also a higher percentage of patients with previous surgery. I think that these aspects prevent us from attributing less blood loss to the laparoscopic technique. There is a selection bias of the cases included in both groups. In this sense, it would be recommendable that the authors established their current criteria for indicating open surgery in this clinical context.
-It cannot be affirmed that there are no differences in the long-term prognosis since the patients included in the laparoscopic and open surgery groups present different indications. 50% of the patients in the open surgery group were operated on by a CCC. I understand that it is a cholangiocarcinoma, although the abbreviation is not defined in the article. If I am correct, was a hepatic hilum lymphadenectomy performed in these cases?
-The absence of differences in hospital stay is also striking, which is one of the benefits expected from laparoscopic surgery. This aspect is not mentioned in the discussion.
In addition to these issues that I consider to be more important, there are other minor aspects that in my opinion could be useful to improve the study.
-The exclusion criteria of 4 or more lesions for laparoscopic surgery has currently been maintained.
-Has the % of cases treated with laparoscopic surgery changed during the inclusion period?
I have noticed some errors in the placement of the data in tables 1, 3, 5 and 7.
